# The Influence of a Polyphenol-Rich Red Berry Fruit Juice on Recovery Process and Leg Strength Capacity after Six Days of Intensive Endurance Exercise in Recreational Endurance Athletes

**DOI:** 10.3390/nu16101428

**Published:** 2024-05-09

**Authors:** Sarah Valder, Elisabeth Habersatter, Tihomir Kostov, Sina Quenzer, Lukas Herzig, Jakob von Bernuth, Lynn Matits, Volker Herdegen, Patrick Diel, Eduard Isenmann

**Affiliations:** 1Department of Preventive and Rehabilitative Sport Medicine, Institute of Cardiovascular Research and Sport Medicine, German Sport University Cologne, 50933 Cologne, Germany; 2Department of Molecular and Cellular Sport Medicine, Institute for Cardiovascular Research and Sport Medicine, German Sport University Cologne, 50933 Cologne, Germany; 3Eckes-Granini Group GmbH, 55268 Nieder-Olm, Germany; 4Department of Beverage Research, Chair Analysis and Technology of Plant-Based Foods, Geisenheim University, 65366 Geisenheim, Germany; 5Clinical & Biological Psychology, Institute of Psychology and Education, Ulm University, 89081 Ulm, Germany; 6Division of Sports and Rehabilitation Medicine, Department of Medicine, Ulm University Hospital, 89081 Ulm, Germany

**Keywords:** polyphenol-rich fruit juice, HIIT, muscle damage, oxidative status, leg strength capacity

## Abstract

Background: Various nutritional strategies are increasingly used in sports to reduce oxidative stress and promote recovery. Chokeberry is rich in polyphenols and can reduce oxidative stress. Consequently, chokeberry juices and mixed juices with chokeberry content are increasingly used in sports. However, the data are very limited. Therefore, this study investigates the effects of the short-term supplementation of a red fruit juice drink with chokeberry content or a placebo on muscle damage, oxidative status, and leg strength during a six-day intense endurance protocol. Methods: Eighteen recreational endurance athletes participated in a cross-over high intensity interval training (HIIT) design, receiving either juice or a placebo. Baseline and post-exercise assessments included blood samples, anthropometric data, and leg strength measurements. Results: A significant increase was measured in muscle damage following the endurance protocol in all participants (∆ CK juice: 117.12 ± 191.75 U/L, ∆ CK placebo: 164.35 ± 267.00 U/L; *p* = 0.001, η^2^ = 0.17). No group effects were detected in exercise-induced muscle damage (*p* = 0.371, η^2^ = 0.010) and oxidative status (*p* = 0.632, η^2^ = 0.000). The reduction in strength was stronger in the placebo group, but group effects are missing statistical significance (∆ e1RM juice: 1.34 ± 9.26 kg, ∆ e1RM placebo: −3.33 ± 11.49 kg; *p* = 0.988, η^2^ = 0.000). Conclusion: Although a reduction in strength can be interpreted for the placebo treatment, no statistically significant influence of chokeberry could be determined. It appears that potential effects may only occur with prolonged application and a higher content of polyphenols, but further research is needed to confirm this.

## 1. Introduction

Physical activity can lead to a shift in physiological homeostasis. Depending on the exercise intensity, volume, and load, exercise-induced muscle damage (EIMD), inflammation, and reactive oxygen species (ROS) accumulation can occur [1,2,3,4]. High-intensity endurance training, in particular, can lead to a shift in physiological homeostasis, resulting in reduced performance [5,6]. Nutritional strategies are used to restore homeostasis as quickly as possible. One strategy that has been discussed for a while is the use of polyphenol-rich foods [7]. Research suggests that chokeberry juice and mixed juices with chokeberry content may be a promising source of polyphenols for athletes due to their high levels of phenolic acids, proanthocyanidins, anthocyanins, flavonols, and flavanones [8]. Furthermore, chokeberry juice shows the strongest antioxidant properties compared to other polyphenol-rich beverages, with Trolox equivalence antioxidant capacity (TEAC) values that are four times higher than those of blueberry juice, cranberry juice, or red wine [9]. To combat the effects of increased oxidative stress, reduced performance, and loss of strength, some experts recommend taking polyphenol supplements or fruit juices [10,11].

Initial studies [11] determined that the consumption of polyphenol-rich juices improves the regeneration of skeletal muscles by up to 13% and can reduce muscle soreness by up to 29%. Fuster-Muñoz et al. [12] observed similar effects in endurance-based athletes (n = 10) who consumed 200 mL/day of pure polyphenol-rich pomegranate juice in comparison to 200 mL pomegranate juice diluted 1:1 with water (n = 10) and a control group not supplementing pomegranate juice (n = 10) for over 21 days. They found that subjects who consumed pure fruit juice exhibited lower concentrations of oxidative stress, as measured by carbonyl compounds and malondialdehyde (MDA), than those who consumed diluted pomegranate juice or an isoenergetic seasonal piece of fruit in the control group. Similar beneficial effects were also observed in studies with chokeberry. Stevanović et al. found that the acute consumption of 200 mL of pure chokeberry juice (with a total phenol content of 1296.8 mg) in comparison to 200 mL of placebo juice (identical in composition without polyphenols) consumed with a standardized breakfast before a half marathon supported a significant reduction in exercise-induced platelet activity, as measured by platelet–monocyte and platelet–neutrophil aggregates in 10 healthy male runners. This reduction was related to the increased polyphenol intake [13]. Further positive effects on cytokine profiles and total antioxidant capacity (TAC) were shown in elite polish rowers who consumed 150 mL of chokeberry juice (anthocyanin content 23 and 24 mg/mL, respectively) in three portions of 50 mL daily for four and eight weeks in training camps in comparison to placebo supplementation [14,15]. During the four-week training camp, participants (n = 9) in the juice intervention group displayed markedly lower glutathione peroxidase activity after 1 min and significantly reduced superoxide dismutase activity 24 h following the 2000 m rowing ergometer test in comparison to the placebo group (n = 10) [14]. After juice intake in the eight-week training camp, the rowers (n = 10) exhibited lower post-exercise levels of tumor necrosis factor alpha (TNF-α). Additionally, they exhibited significantly higher total antioxidant capacities (TACs) following a 2000 m rowing ergometer test at the end of the eight-week training camp in comparison to the placebo group (n = 9) [15]. These findings suggest an improved oxidative stress status in the juice group compared to other participants. This can be attributed to the increased intake of anthocyanins and flavonoids, which have antioxidant and, in particular, flavonoid scavenger properties, inactivating free radicals [14,15]. However, it is known that oxidative stress is also necessary for muscular adaptations. Sorrenti et al. [16] point out that polyphenols should be ingested before or after a certain period of physical activity and not immediately afterward, to avoid completely suppressing exercise-induced inflammatory processes and inhibiting possible adaptation processes.

In contrast to previous studies, no data are available on the short-term repeated application of chokeberry juice or a red fruit juice drink with chokeberry content during an intensive endurance running training week. Therefore, a commercially available and tasty red fruit juice drink containing 25% chokeberry, 64% red grape, 7% passion fruit, and 2% lemon juice was used. The juice was rated as very palatable in a sensory evaluation. It did not have the same astringent effect as pure chokeberry juice and was, therefore, more suitable as a sports drink. This study aimed to investigate whether a red fruit juice drink with a chokeberry content of 25% and the associated polyphenol intake could already be an adequate natural sports beverage with an impact on exercise-induced muscle damage as well as strength in the lower body. It is also assumed that the ingredients of the red fruit juice drink could influence oxidative stress after the intensive training week.

## 2. Materials and Methods

### 2.1. Participants 

To determine the sample size, a power analysis (F-tests, ANOVA fixed effects, special, main effects, and interaction) was performed a priori. For the calculation, a medium effect (f) (0.25), an α-error of 0.05, and a power of 0.8 (1–β error) were specified. Based on the study design and the two-arm crossover model (df = 1), a sample size of 17 subjects (34 in total) was calculated. The power analysis and the study design are based on previous studies [17,18,19]. The study was successfully completed with 18 subjects, comprising 4 women and 14 men. Figure 1 provides an overview of the recruitment process and the reasons for drop-out.

All participants of the study were recruited from local running groups and endurance courses of the German Sport University Cologne.

They were healthy and generally physically active but not trained specifically. In addition, subjects had to be between 18 and 35 years old, not consume a vegan diet, be non-smokers, have no acute injuries or chronic illnesses, not take any medication, and have a running time for 10,000 m of <55 min for women and <50 min for men. High performance athletes with training volumes of >100 km × wk^−1^ and individuals trained specifically for national and international competitions with >10 training sessions × wk^−1^ or >10 h training time × wk^−1^ were also excluded [20,21,22].

Before the subjects were included in the study, they were informed about the study design and objectives and had to give their written consent to participate. All personal data collected were anonymized in compliance with the data protection regulations in Germany.

The entire study design was approved by the local ethics committee of the German Sports University Cologne in August 2022 (Ethics number: 132/2022, date of approval: 22 August 2022) and is in accordance with the Declaration of Helsinki. 

### 2.2. Study Design 

The study was designed as a randomized, double-blind trial with a crossover design. To ensure accuracy, there was a washout period of 4 weeks between intervention phases to account for possible training adaptations. The intervention phase, which included a six-day endurance training program, was carried out twice with juice and placebo supplementation. The supplementation intervals were split into a three-day familiarization phase and a six-day intervention phase. Throughout the study, all participants used a nutrition documentation app to record their dietary intake. At the start and finish of each training phase, an initial and baseline examination was conducted, including taking a blood sample, determining anthropometric data such as body weight and height, and measuring leg strength capacity using a velocity-based deep back squat method [23]. Figure 2 displays the entire study design.

### 2.3. Study Beverages

As part of the study, participants were instructed to consume 200 mL of a red fruit juice drink or a placebo every morning and evening after a meal. These drinks were provided by the Eckes-Granini Group. The red fruit juice contained a minimum of 98% fruit juice content, with 64% red grape, 25% chokeberry, 7% passion fruit, and 2% lemon juice, according to the label. The placebo used was an iso-caloric, dark-colored drink that closely resembled the fruit juice in color, taste, and potassium content. However, the polyphenol content of the placebo was minimized to a level of 91 mg/L, in contrast to the 4105.5 mg/L on average found in the red fruit juice (see Appendix A). To distinguish between placebo and fruit juice, the bottles were coded differently and blinded. Only an independent person could later assign the study drinks. The produced drinks were maintained throughout the full storage period at 18 °C in a dark storage room.

### 2.4. Analysis of Phenolic Ingredients

The UV-VIS spectrophotometer (PerkinElmer, Lambda 365, Waltham, MA, USA (serial number 365K9120205)) was used to determine the total polyphenol content of the raw materials. These measurements were performed using Folin–Ciocalteu reagents (VWR, Darmstadt, Germany, Art. No. ICNA19518690).

The total phenolic content was determined as the equivalent amount of catechin. To establish the linear calibration line, 212.4 mg catechin hydrate was pre-dissolved in a 100 mL volumetric flask with a few milliliters of methanol and then filled up to 100 mL with deionized water. Catechin solutions of 40 mg/L, 200 mg/L, 400 mg/L, 600 mg/L, and 1000 mg/L were then prepared from the stock solution.

To measure the samples, 8.4 mL of deionized water was added using a dispenser. Then, 100 μL of the sample and 500 μL of Folin–Ciocalteu reagent were added, and the solution was shaken using a test tube shaker. For the reagent blank, 100 μL of deionized water was added instead of 100 μL of sample. After a short reaction time of approximately three minutes, 1000 μL of sodium carbonate solution was added to each test tube and mixed well. In the presence of phenolic compounds, a reaction is visible after a short time, as a blue color complex is formed. After one hour, the samples are read at 720 nm against the reagent blank. The ingredients and important measurement parameters of the analyses described can be found in Appendix A.

### 2.5. Study Day Schedule

All participants arrived at the morning examination after being well-rested without exercise for 48 h and fasting without a meal for 12 h. Anthropometric measurements for bodyweight and height were taken (seca model 2741900099), and a blood sample was collected (T0). After finishing the first part, a warm-up program was initiated according to the NSCA guidelines [24]. Prior to the study, participants were familiarized with the methodology and execution of the squat. In order to accomplish this, three sets of ten repetitions without additional weight on the squat bar were performed twice, each with a rest interval of 5 min. A speed-based approach was selected to minimize the risk of injury to the musculature during testing for inexperienced weightlifting participants [23]. The Speed4Lift velocity sensor (SL4, Madrid, Spain) was used to record velocity at defined load levels to ensure accurate measurement. This indirect one-repetition maximum (1RM) test commenced with the first load level set at approximately 30% of the participant’s self-estimated 1RM. If the 1RM could not be estimated, the first load level was set at 40% of the body weight. Subsequently, additional load levels were introduced at an increase of 20% of the self-proclaimed 1RM or 20% of body weight, respectively, until the speed reached ≤0.6 m per s (70% of 1RM and cut off-point for the high reliability of the data [23]). Assuming a linear regression of the force–velocity relation and a mean velocity of the 1RM of the squat at 0.2 m per s, the estimated 1RM (e1RM) was determined [23].

### 2.6. Endurance Training Protocol

On the following day, the subjects began the six-day endurance training protocol, which consisted of four high-intensity interval training sessions and two endurance running sessions with moderate intensity. A sequence of interval session, endurance run, interval session, endurance run, and two interval sessions at the end was chosen to meet the general guidelines for a HIIT protocol training design [25]. The training intensity was controlled by the individual lactate threshold calculated by the D-mod method (LT(D)) [26] of the study participants and comprised 120% of the LT(D) in the interval units of 5 × 4 min load (rest 2 min of low-intensity running) and 65% of the LT(D) in the endurance runs. Intensity, number of repetitions, and duration of the interval units, as well as the intensity and duration of the endurance runs, were again set according to the guidelines for a HIIT training design [25]. The LT(D) was determined subject-specifically before the study via a field-level test. In addition, each session always included a warm-up and cool-down phase (low-intensity running) as well as a stretching or agility program before the interval sessions. The documentation of the training process of the study participants was carried out using subject-specific apps (Apple Fitness+, Apple Distribution International Ltd., Cupertino, CA, USA; or Strava, Strava, Inc., San Fransisco, CA, USA). The complete training plan can be found in Figure 3. After the completion of the training protocol, all subjects reappeared the following day fasted (12 h without a meal) in the morning for baseline testing with the repeated collection of anthropometric data, a blood draw (T1), and for the measurement of leg strength using the deep back squat according to the methodology described in the previous section.

### 2.7. Blood Analyses

In order to assess the designated level of training intensity, skeletal muscle-specific creatine kinase (CK) concentration in serum was determined as a marker using a COBAS h 232 point-of-care system (Roche Diagnostic Systems, Rotkreuz, Switzerland). 

For the evaluation of the endurance exercise-induced oxidative stress status, oxidized low-density lipoprotein (oxLDL) serum concentrations were measured using the Human oxLDL Adduct Elisa Kit (Bensheim, Germany).

Serum concentrations of interleukins (IL) 6 and 10 were analyzed via Human IL-6 and Human IL-10 Quantikine ELISA Kits (R&D Systems, Minneapolis, MN, USA) to monitor modifications in pro- and anti-inflammatory responses related to endurance exercise training as well as the intake of the polyphenol-rich red fruit juice. 

### 2.8. Nutrition Records

Dietary logging for the duration of the two intervention intervals was performed using the Food Diary Food Database (FDDB) Extender (Food Database GmbH, 28217 Bremen, Germany). The subjects were instructed to keep their diet constant over the two intervention periods and to consume the same foods in each case. Participants were asked to abstain from alcohol and very polyphenol-rich foods such as coffee, fruit juices, cocoa or chocolate, and black and green tea during intervention periods. The food protocols were then transferred to EBISpro 2016 in order to cover a wider range of nutrients. 

### 2.9. Statistical Analyses

The data were analyzed using R version 4.3.0 [27]. The results are presented as means and standard deviations (SD). All parameters underwent a Z transformation at each time point to detect and assess potential outliers. Values exceeding a triple SD were excluded from the analysis. The parameters of the endurance training protocol (training time, training km, endurance km, interval km, and pace/total) as well as the nutritional data (energy, carbohydrate, fat, protein, EAA, and leucine) were analyzed using a paired *t*-test. Q–Q plots, histograms, and the Shapiro–Wilk test were used to test for a normal distribution of the residuals. In instances where data did not conform to a normal distribution (energy and carbohydrate intake per day), the Wilcoxon test, suitable for non-normally distributed datasets, was employed. The raw data of CK, oxLDL, IL-6, and IL-10 were transformed to their decadic logarithm (log10) before data analysis. The transformed data are additionally presented as means and standard error (SE) in Appendix A. Linear mixed-effects regression models were used with time (before vs. after) and group (juice vs. placebo) and their interaction (time × group) as predictors for the effects of juice on training protocol (R package lme4; [28]). 

The Akaike Information Criterion (AIC) and the Bayesian Information Criterion (BIC) were employed to choose the linear mixed-effects regression model that strikes the optimal balance between model fit and complexity. This selection was based on evaluating minimum values in relation to the likelihood function, aiming to identify the model with the highest probability for an effective fit while considering its overall complexity. Due to the high intraclass correlation (ICC = [0.310; 0.792]), random incepts were assumed. Where necessary (assessed via Chi^2^ test, AIC, BIC), a random effect for the intervention was assumed (oxLDL, IL-6, IL-10, e1RM, relative e1RM (rel. e1RM)). Shapiro–Wilk tests, Q–Q plots, and histograms were used to determine the normality and homoscedasticity of the model residuals. To review the robustness of the results, we repeated the analyses adjusting for possible covariates of sex, age, and BMI with no further influence on the results. (see Appendix A). As effect size partial η^2^ is reported for time effects, group effects, and time × group interaction. All reported *p*-values were two-tailed with a significance level of α < 0.050. 

## 3. Results

The 18 participants consistently followed the study guidelines throughout the research period, including the prescribed beverage intervention and exercise plan where feasible. In the analysis of CK, one subject could not be included due to exceptionally high CK levels at baseline, surpassing values beyond a triple SD in the Z-transformed data. All anthropometric data are shown in Table 1.

In both study arms, all participants completed the intensive endurance training protocol with 210.00 ± 00.00 min of exercise and a total of 47.76 ± 9.34 km in the juice condition and 47.40 ± 9.33 km in the placebo condition. With 27.67 ± 7.24 km in the juice treatment and 26.66 ± 7.42 km in the placebo treatment, the kilometers run during the interval sessions were slightly higher than the kilometers run during the endurance run sessions (juice: 20.09 ± 2.49 km; placebo: 20.74 ± 2.40 km). The mean running speed was almost identical for both intervention drinks, at 4.58 ± 0.84 min/km for juice and 4.58 ± 0.81 min/km for placebo. Therefore, there were no statistically significant differences between the two study drinks in any of the training parameters (see Table 2). Individual values can be found in Appendix A. 

### 3.1. Effects of Six-Day Running and Juice on Skeletal Muscle Damage—Creatine Kinase

To determine the effectiveness of the six-day endurance training program, the concentration of CK in the serum was measured to assess the status of muscle damage.

After the six-day repeated training load (T1 (+7d)), in both groups, CK serum concentrations increased significantly over time (*p* ≤ 0.001, η^2^ = 0.170, Figure 4). However, there was no significant group effect (*p* = 0.371, η^2^ = 0.010), and there was also no significant group * time interaction at T1 (post) (*p* = 0.976, η^2^ = 0.000).

### 3.2. Effects of Six-Day Running and Juice on oxLDL 

The concentrations of oxLDL, as a marker of oxidative stress, were analyzed in serum at T0 and T1 (+7d) to investigate the effect of training and juice and their possible interactions on endogenous oxidative stress level (Figure 5). No significant changes were detected over time between T0 and T1 (+7d) (*p* = 0.120, η^2^ = 0.040). There were also no significant differences between the groups (*p* = 0.632, η^2^ = 0.000), and no significant interactions between the group and time were observed (*p* = 0.354, η^2^ = 0.010).

### 3.3. Effects of Juice on Recovery Processes and Strength Capacity following Six-Day Running

To analyze the connection between oxidative stress (related to physical activity), inflammatory responses, and beverage intakes, the serum concentrations of the pro-inflammatory cytokine IL-6 and the anti-inflammatory cytokine IL-10 were determined at T0 and T1 (+7d) (Figure 6).

After the six-day repeated training load, there was no significant change observed in time and group for IL-6 (time: *p* = 0.256, η^2^ = 0.040; group: *p* = 0.383, η^2^ = 0.020; Figure 6A). Likewise, no significant effects were observed for IL-10 (time: *p* = 0.737, η^2^ = 0.000; group: *p* = 0.434, η^2^ = 0.010; Figure 6B). No time group effect was detected at T1 for either parameter (IL-6: *p* = 0.278, η^2^ = 0.030; IL-10: *p* = 0.189, η^2^ = 0.040).

### 3.4. Effects of Six-Day Running and Juice on e1RM and relative e1RM in Back Squat

After six days of training, in the juice intervention, leg strength increased from 104.22 ± 24.21 kg to 105.56 ± 24.54 kg. In contrast, leg strength decreased from 104.28 ± 22.00 kg to 100.95 ± 17.74 kg when the placebo was consumed (Figure 7A). However, this difference was not statistically significant for time (*p* = 0.168, η^2^ = 0.030), group (*p* = 0.988, η^2^ = 0.000), and time × group interaction (*p* = 0.172, η^2^ = 0.030). Comparable results are shown for the relative e1RM with no significant effects for time (*p* = 0.438, η^2^ = 0.010), group (*p* = 0.941, η^2^ = 0.000), and time × group interaction (*p* = 0.269, η^2^ = 0.000, Figure 7B).

All results are summarized in Appendix A.

### 3.5. Evaluation and Comparison of the Nutrient Intake during the Fruit Juice Intervention and Placebo Consumption Phase

The intake of energy, carbohydrates, leucine, and EAA was measured during the juice intervention and placebo consumption phase.

No statistically significant variance was detected in energy, fat, protein, leucine, and essential amino acid (EAA) intake per day between the cases of consuming juice and placebo. However, carbohydrate intake per day was marginally higher during the placebo consumption phase (refer to Table 3). Unfortunately, it was impossible to determine the actual intake of polyphenols with EBISpro.

## 4. Discussion

This study aimed to investigate the influence of a short-term application of a polyphenol-rich red fruit juice drink as a naturally based sports beverage with a chokeberry content of 25% during an intense six-day endurance exercise protocol on training-induced skeletal muscle damage (CK serum concentration), oxidative status (oxLDL), inflammatory processes (IL-6 and IL-10), and strength capacity in recreational athletes.

The results clearly indicate that the designed endurance training protocol significantly increased CK serum concentration (*p* = 0.001, η^2^ = 0.17) without sex differences, and there was repetitive loading training stimulation. However, the intake of the study beverages showed no significant effects on CK concentration. These observations are in line with previous studies investigating the effects of strength training [29] or marathon running [30] on serum CK concentration without differences between a polyphenol-rich Montmorency Cherry Juice and a placebo intake [29,30]. Conversely, Pilaczynska-Szczesniak et al. [14] found significantly lower serum CK concentrations after aronia juice application only one min after a 2000 m rowing ergometer test. In contrast to Pilaczynska-Szczesniak et al. [14], the CK concentration was not measured acutely after exercise but 24 h after exercise, which can lead to different results. It is known that the CK concentration can be influenced by various factors, such as performance level or training protocol, and that the peak concentration occurs between 24 and 96 h after exercise [31].Concerning oxidative stress, similar to Nieman et al. [32], no significant difference in oxLDL concentration was found after exercise or between the treatments. In contrast, previous studies have shown a reduction in oxLDL concentration immediately after exercise [33,34]. It appears that the time of blood sampling in this study and in Nieman et al. [32] was chosen too far away from the last exercise so that no change could be detected. However, initial studies also show that continuous physical activity can also reduce the oxLDL concentration [33,34,35]. Consequently, no statement can be made regarding whether a polyphenol-rich red berry fruit juice can reduce training-induced oxidative stress. In context, it appears that the performance level and the exercise protocol influence the change in oxLDL concentrations. Studies indicate that marathon runners with low VO2max values have higher oxLDL concentrations and that intensive exercise induces a higher increase [36,37]. 

Similar to oxidative stress, no significant time or treatment effect could be observed for IL-6 and IL-10.

This is consistent with the review by Rickards et al. [11], who found no effects on IL-6 of polyphenol-containing foods, juices, and concentrates. However, it has already been shown that consumption of tart cherry juice, which contains polyphenols, can attenuate cytokine concentrations of the pro-inflammatory IL-6 during prolonged endurance exercise such as a marathon or simulated cycle race [30,38]. The controversial results may be due to the significantly higher serum IL-6 concentrations after a marathon (juice: 41.8 pg/mL vs. placebo: 82.1 pg/mL) [30] compared to the present IL-6 concentrations after the six-day intensive endurance training (juice: 0.74 ± 0.64 pg/mL; placebo: 1.09 ± 1.54 pg/mL). A similar assumption can be made for IL-10, as combined curcumin and pomegranate extract supplementation (1000 mg/d) before (26 d) and after (4 h and 24 h) a half marathon race resulted in significantly higher serum concentrations of the anti-inflammatory cytokine [39]. Additionally, the lack of statistical significance in the cytokine outcomes of this study might be attributed to the polyphenolic composition present in the fruit juice. The commercially available and naturally based red fruit juice drink used in this study contained only 25% chokeberry and, therefore, 821.1 mg polyphenols/200 mL (bottle). Compared to 200 mL of pure chokeberry juice used in the study by Stevanović et al. [13], this is 475.7 mg less polyphenols. It should be emphasized that pure chokeberry juice exhibits a high potassium content of 1969 mg/L [9], while the red fruit juice in our study averages only 1448 mg/L, indicating a difference of 521 mg/L. This variance in potassium intake from pure chokeberry juice might contribute to its beneficial effects. The supplementary potassium could potentially compensate for losses through sweating or potassium ion flux from the muscle membrane during depolarization in neuronal action potentials, thereby promoting muscle regeneration [40].

As a novel approach, this study integrated both molecular and functional parameters to investigate the interplay between muscle damage, polyphenol intake, and subsequent strength loss, a dimension less explored in previous research [41]. 

As shown in Figure 7, the absolute values of e1RM in the deep squat indicate a positive effect of juice compared to placebo (placebo: ∆ −3.33 ± 11.49; juice: ∆ 1.34 ± 9.26). The absolute loss of strength in the placebo group is similar to previous studies [19]. However, no significant differences over time and treatment were found. This could be due to the velocity sensor methodology used. Even though the e1RM measurement using speed sensors is a valid method, it has some limitations. A decisive factor is the larger standard deviation than when determining the actual maximum force [23]. The study of Hooper et al. [42] also showed no statistically significant results for the consumption of 500 mg of a polyphenol-rich tart cherry juice extract for 7 days in terms of countermovement jump (CMJ) power in 13 men performing six sets of 10 repetitions of squats compared to a placebo. Similar to the results of the study presented here, jump height trended towards significance for the condition of polyphenol intake. As mentioned in the previous section, pure chokeberry juice may have better effects due to its higher potassium content. Recent study results show that low muscle glycogen levels and low intramuscular potassium ion concentrations can contribute to muscle fatigue and loss of performance [43]. 

Our study participants largely adhered to our directive of abstaining from particularly polyphenol-rich foods and maintaining a consistent diet during both intervention phases. This is supported by the fact that the intake of energy, fat, protein, essential amino acids, and leucine did not significantly differ between the juice and placebo intervention phases [44]. Leucine, found abundantly in high-quality protein sources, is a branched-chain essential amino acid that fuels protein synthesis and curbs protein breakdown. It is, therefore, considered highly relevant for muscle protein synthesis and regeneration [45,46].

While carbohydrate intake was on average ten grams higher during the placebo phase (refer to Table 3), this discrepancy is minimal and likely irrelevant. To put these results into perspective, ten grams of carbohydrates roughly equate to half a banana or two rice cakes. Taken together, no significant differences in nutrient or energy intake were observed that could account for an impact on strength capacity. 

### Limitations/Outlook

This study introduces novel insights while acknowledging practical limitations inherent in its execution. These should be taken into account when interpreting the results.

OxLDL as a marker of oxidative and exercise-induced stress status showed no statistically significant results, probably because the time of measurement was inadequate. Other studies measure over a window of time from the immediate post-exercise period up to 72 h after exercise [32,35,36]. Changes due to intensive endurance training and juice consumption were probably not recorded in this study because measurements were taken at only one point in time. In addition, the measurement of oxidative stress should be complemented by other markers such as total antioxidant capacity (TAC), glutathione peroxidase, and superoxide dismutase activity, as these have been shown as antioxidant effects of chokeberry juice in studies of intensive endurance exercise [14,15] and are considered to be established markers of the oxidative stress state in exercise [47]. 

Another limitation is the velocity-based deep back squat method which can only estimate the 1RM and potentially does not produce statistically significant results in the strength test. However, it should be noted that the speed-based approach used in this study has an acceptable level of accuracy compared to conventional 1RM tests, which has been scientifically validated [48].

In addition, the definition of intensity as a percentage of LT(D) in the six-day endurance training protocol can be seen as a limitation, as effort-matched approaches are considered to be a more realistic way for athletes to approach their training [6,49]. However, the training sessions in this study could only be carried out if the subjects were able to train independently according to the defined protocol. In comparison to studies with older people [50], we cannot be sure that the training has been in accordance with the specifications down to the last detail, but this study shows that training tracking works with younger people and gives valid results.

Another potential limitation is the reliance on self-reported data regarding food intake, which may not consistently align with the participants’ actual dietary habits. Regrettably, the assessment using EBISpro did not allow for the evaluation of polyphenol intake.

However, as shown in the European Prospective Investigation into Cancer and Nutrition (EPIC) study, Europeans consume, on average, 1.2 g of polyphenols per day, and the main food sources are coffee, tea, fruits, and wine [51]. Considering that our participants were instructed to avoid these food groups during the period of the study, we assume that an additional intake of about 1.6 g of polyphenols per day has significantly augmented the overall polyphenol consumption during the fruit juice intervention phase. More polyphenols could have been delivered if a higher proportion of chokeberry or pure chokeberry juice had been used in this study. However, this would have made the juice more unpleasant to taste. The sour, acidic, and astringent taste of chokeberry and chokeberry juice usually limits their use in pure form. However, mixed drinks containing chokeberry juice are already produced industrially and are accepted and consumed by consumers [9]. Finally, this study was able to show that a chokeberry content of 25% in the commercial red fruit juice drink is too low to achieve the effects already shown in studies with pure chokeberry juice. Therefore, for a natural fruit juice-based ergogenic sport beverage, the chokeberry content needs to be increased while maintaining the taste of the beverage.

## 5. Conclusions

This study explored the impact of a short-term polyphenol-rich red fruit juice drink or placebo on athletes undergoing a six-day endurance training protocol. While the designed training induced significant muscle damage, the study beverage showed no significant effects on exercise-induced muscle damage or oxLDL concentrations, consistent with findings in similar exercise interventions. The investigation integrated molecular and functional parameters, shedding light on the interplay between muscle damage and strength loss, an aspect less explored in previous research. Although the absolute values of e1RM measured in the deep squat indicate a positive effect of the juice, these are not significant. In future studies, the potential positive effect of polyphenol-rich juice on leg strength should be investigated using a more precise methodology. Despite limitations in the timing of oxLDL measurement and the reliance on self-reported dietary data, our study provides valuable insights into the complex interactions of polyphenol intake and exercise on athletes’ physiological responses during intensive endurance training, encouraging further exploration and methodological refinement in understanding the impact of polyphenols in athletes.

## Figures and Tables

**Figure 1 nutrients-16-01428-f001:**
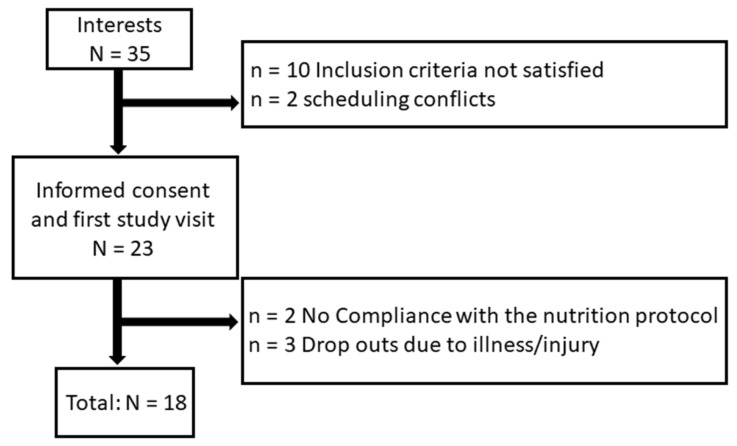
Overview study recruitment.

**Figure 2 nutrients-16-01428-f002:**
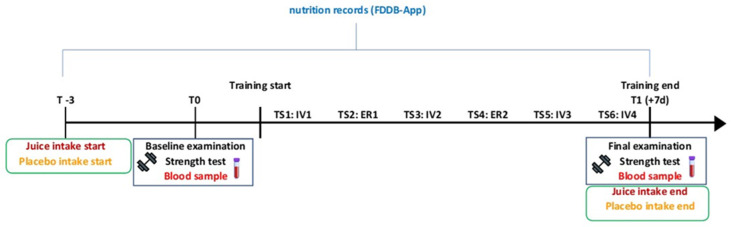
Study design. TS = Training session, IV = Interval training, ER = Endurance run. Pictorial sources: Dumbbell—Free sports icons (flaticon.com).

**Figure 3 nutrients-16-01428-f003:**
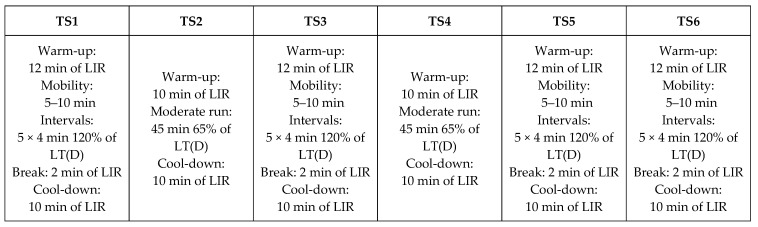
Six-day training plan consisting of four interval units with 120% of the LT(D) and two endurance runs with 65% of the LT(D) as load intensity specification (accompanying warm-up and cool-down phase included). LT(D) = lactate threshold calculated by the D-mod method. LIR = low intensity running.

**Figure 4 nutrients-16-01428-f004:**
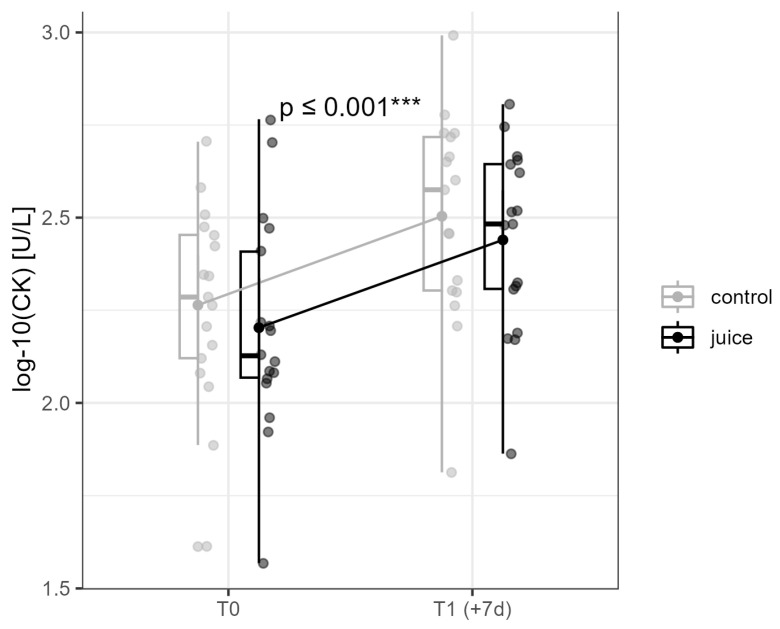
Interaction diagram for CK in the control and juice intervention (before and after an endurance exercise interval of 6 days). CK = creatine kinase. *** ≤ 0.001.

**Figure 5 nutrients-16-01428-f005:**
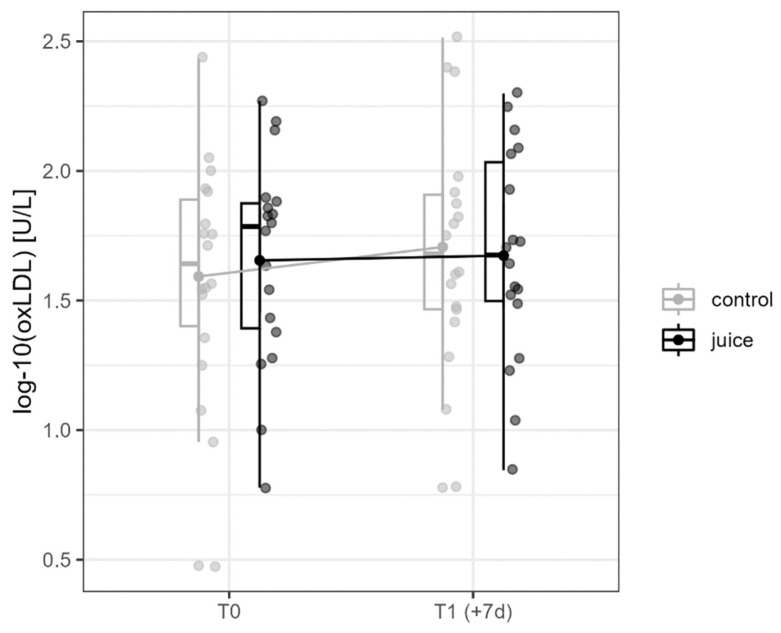
Interaction diagram for oxLDL in the control and juice intervention (before and after an endurance exercise interval of 6 days). oxLDL = oxidized low-density lipoprotein.

**Figure 6 nutrients-16-01428-f006:**
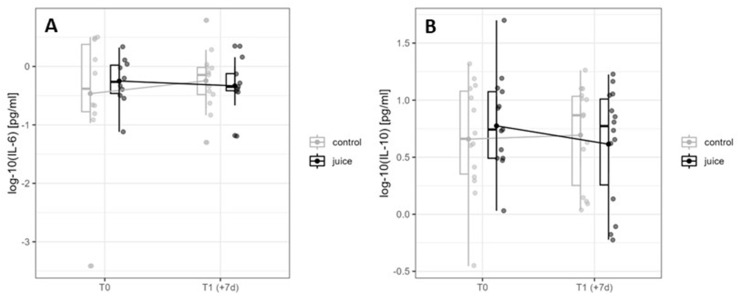
Interaction diagram for IL-6 (**A**) and IL-10 (**B**) in the control and juice intervention (before and after an endurance exercise interval of 6 days). IL-6 = Interleukin 6. IL-10 = Interleukin 10.

**Figure 7 nutrients-16-01428-f007:**
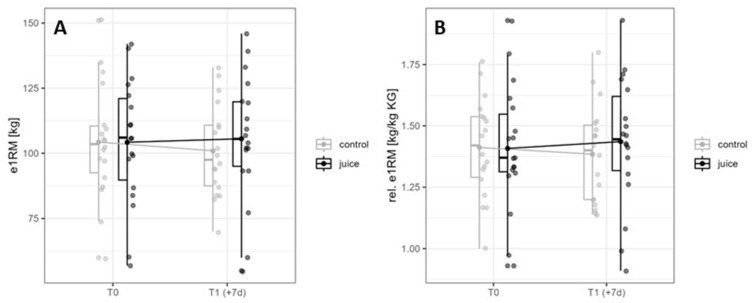
Interaction diagram for e1RM (**A**) and relative e1RM (**B**) in the control and juice intervention (before and after an endurance exercise interval of 6 days). e1RM = estimated 1RM. rel. e1RM = relative e1RM.

**Table 1 nutrients-16-01428-t001:** Overview of subject characteristics.

Sex	Men (14), Women (4)
Age [years]	24.5 ± 3.1
Height [m]	1.79 ± 0.06
Body weight [kg]	73.10 ± 8.30
BMI [kg/m^2^]	22.84 ± 1.74

BMI = body mass index.

**Table 2 nutrients-16-01428-t002:** Six-day endurance training protocol documentation for the study conditions of juice and placebo.

Parameter	Juice	Placebo	*p*-Value
Training time [min]	210.00 ± 00.00	210.00 ± 00.00	---
Training km [km]	47.76 ± 9.34	47.40 ± 9.33	0.654
ER km [km]	20.09 ± 2.49	20.74 ± 2.40	0.094
Interval [km]	27.67 ± 7.24	26.66 ± 7.42	0.075
Pace/total [min/km]	4.58 ± 0.84	4.58 ± 0.81	0.981

ER = endurance run.

**Table 3 nutrients-16-01428-t003:** Intake of energy, carbohydrates, leucine, and EAA during juice intervention and placebo consumption phase.

	Juice Group	Placebo Group	*p*-Value
Energy [kcal/d]	2598.12 ± 723.73	2650.14 ± 739.60	0.163
Carbohydrates [g/d] % of Energy	321.79 ± 91.19 49.82 ± 6.51%	330.79 ± 97.09 51.00 ± 4.97%	0.044
Fat [g/d] % of Energy	96.50 ± 30.56 32.88 ± 4.11%	97.32 ± 30.01 32.58 ± 4.20	0.469
Protein [g/d] % of Energy	103.87 ± 8.43 16.41 ± 2.85%	105.76 ± 8.347 16.41 ± 2.76%	0.095
Leucine [g/d]	8.38 ± 2.86	8.53 ± 2.83	0.101
EAA [g/d]	50.95 ± 17.139	51.85 ± 17.00	0.099

EAA = Essential amino acids.

## Data Availability

The data presented in this study are available on request from the corresponding author. The data are not publicly available due to privacy.

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
