# Peer review of "The Influence of a Polyphenol-Rich Red Berry Fruit Juice on Recovery Process and Leg Strength Capacity after Six Days of Intensive Endurance Exercise in Recreational Endurance Athletes"

_nutrients, 2024, doi:10.3390/nu16101428_

Round 1

Reviewer 1 Report

Comments and Suggestions for Authors

This study investigates the effects of short-term supplementation of a chokeberry juice or a placebo on muscle damage, oxidative status, and leg strength during a six-day intense endurance protocol. The manuscript introduces novel insights, but the data are very limited. I have the following questions for the authors:

1. Why not just take polyphenol supplements instead of fruit juices?

2. The measurement of oxidative stress should be supplemented with other indicators.

3. Change every “Error! Reference source not found” to the corresponding figure or table.

4. The applicable population range of this study is too narrow, and further research can be conducted on participants of different age.

5. Where are the Table S1, S2 and S3?

6. Check if “ml” and “l” in the article should be changed to “mL” and “L”. If so, please write it uniformly.

7. Why was the polyphenol content of the placebo 91 mg/L, and the 4105.5 mg/L on average in the red fruit juice?

8. This study proposed many limitations, how to solve these problems?

Author Response

We would like to thank both reviewers again for their important and critical comments and hope that our changes meet their expectations.

Reviewer 2 Report

Comments and Suggestions for Authors

The aim of this study is to investigate whether a red fruit juice drink with a chokeberry content of 25 % and the associated polyphenol intake could already have an impact on exercise-induced muscle damage as well as strength in the lower body.

It is an interesting study, with an interesting subject matter but with a very limited methodological approach.

Here are my contributions:

There is a need to organise the information provided in the introduction and to clarify and improve all content related to the research carried out so far. In some studies the amount used is specified, in others it is not... it is necessary to clarify: supplementation taken, amount and how its effect was assessed.

In lines 90, 272, 281… there are an incorrectly inserted references.

The text states that 23 subjects were included in the study but the figure shows 18 as the final number. Wouldn't it be more correct to put 18 in the text? And then due to the diet 17 participants finish.

Line 333, drawing conclusions for women with only 4 subjects is too ambitious. The results and statistical calculations are not separated by gender, so no such conclusions can be drawn.

The study seems designed to test the effects of a commercial product, not the effects of a particular substance. According to the researchers' own comments in the discussion of the manuscript, the concentrations seem not to be the most appropriate for use as an ergogenic supplement, is this the case? What is the point of conducting the study knowing that the concentrations may not be the most appropriate?

To what extent do the researchers believe that the different diets of the subjects participating in the study may have had an influence? Nothing is said in the article about this.

It does not seem sufficient reason to provide a supplement with inadequate concentrations because with adequate concentrations the taste would be worse, does it not?

Don't you consider the sample size and the unequal gender grouping a limitation of the study?

Author Response

(The authors gave the same response as above.)

Round 2

Reviewer 1 Report

Comments and Suggestions for Authors

N.A

Author Response

Thank you very much for your constructive comments.

Reviewer 2 Report

Comments and Suggestions for Authors

Despite considerable improvements to the manuscript, two limitations remain:

- The introduction still needs to be improved.

- The supplementation product used for the research does not meet the sufficient concentrations recommended in the literature, as the authors state in the text, so the research lacks novelty and interest.

Author Response

Thank you very much for your comments. We have revised our manuscript again and also incorporated the editor's comments. We hope that the revised version meets your expectations.
